# Extracellular Vesicles in Sickle Cell Disease: A Promising Tool

**DOI:** 10.3390/bioengineering9090439

**Published:** 2022-09-05

**Authors:** Yann Lamarre, Elie Nader, Philippe Connes, Marc Romana, Yohann Garnier

**Affiliations:** 1Université Paris Cité and Université des Antilles, Inserm, BIGR, F-75015 Paris, France; 2Laboratoire Inter-Universitaire de Biologie de la Motricité EA7424, Team “Vascular Biology and Red Blood Cell”, Université Claude Bernard Lyon 1, Université de Lyon, 69622 Lyon, France

**Keywords:** sickle cell disease, SCD, extracellular vesicles, EVs, microparticles, exosomes, biomarkers, endothelial cells, neutrophils, microRNA

## Abstract

Sickle cell disease (SCD) is the most common hemoglobinopathy worldwide. It is characterized by an impairment of shear stress-mediated vasodilation, a pro-coagulant, and a pro-adhesive state orchestrated among others by the depletion of the vasodilator nitric oxide, by the increased phosphatidylserine exposure and tissue factor expression, and by the increased interactions of erythrocytes with endothelial cells that mediate the overexpression of adhesion molecules such as VCAM-1, respectively. Extracellular vesicles (EVs) have been shown to be novel actors involved in SCD pathophysiological processes. Medium-sized EVs, also called microparticles, which exhibit increased plasma levels in this pathology, were shown to induce the activation of endothelial cells, thereby increasing neutrophil adhesion, a key process potentially leading to the main complication associated with SCD, vaso-occlusive crises (VOCs). Small-sized EVs, also named exosomes, which have also been reported to be overrepresented in SCD, were shown to potentiate interactions between erythrocytes and platelets, and to trigger endothelial monolayer disruption, two processes also known to favor the occurrence of VOCs. In this review we provide an overview of the current knowledge about EVs concentration and role in SCD.

## 1. Introduction

Sickle cell disease (SCD) results from a single nucleotide mutation in the gene coding for β-globin. The homozygous inheritance of this allele, noted β^S^, causes the most frequent form of SCD, which affects 312,000 neonates worldwide per year [1] and is called sickle cell anemia (SCA). In SCA, a mutated hemoglobin is produced, the hemoglobin S (HbS), instead of the normal hemoglobin A. The second most common form of SCD is called HbSC disease and is due to the co-inheritance of the β^S^ mutation, with the β^C^ allele. Co-inheritance of the β^S^ allele with other mutations of the β-globin gene, results in other sickle cell syndromes, among which HbSD^Punjab^, HbS^Oarab^, and HbS-βthalassemia. This recessive genetic disorder is clinically characterized by chronic anemia and frequent painful vaso-occlusive crises (VOCs). Besides VOCs, other complications are associated with this disease, such as acute splenic sequestrations, acute chest syndrome, pulmonary hypertension, osteonecrosis, leg ulcers, stroke, and priapism. Hydroxyurea (HU, or hydroxycarbamide) is the most prescribed drug, whereas L-glutamine, voxelotor, and crizanlizumab, which are authorized in the United States, have not received approval in Europe yet [2]. A promising research area has been opened about one decade ago, related to the utility of extracellular vesicles as diagnostic and prognostic tools, but also as therapeutic targets.

Extracellular vesicles (EVs) are a biomarker and an actor modulating the pathophysiology of SCD. EVs are a heterogenous group of membrane-delimited particles produced by nearly all cell types and detectable in multiple biological fluids including urine, broncho-alveolar lavage fluid, sputum, synovial fluid, ascites, saliva, and plasma [3]. None of these vesicles contain the cellular machinery required for replication. Although detectable in physiological condition, EVs have been detected at high levels in numerous diseases including cardiovascular diseases, atherosclerosis [4], cancer, diabetes, [5] and COVID-19 [6], with further increased levels for most severe cases, and are, therefore, considered as biomarkers [7,8]. Indeed, similar to other tests based on blood samples, EVs quantification and characterization is less invasive than many other techniques used to directly assess biological states. Therefore, EVs could be biomarkers with a clinical utility in determining risk in several diseases, including SCD [9]. Originally described as cell dust [10], it has been shown that EVs play a role in cell-to-cell communication at both paracrine and systemic levels [11,12]. Indeed, EVs can carry biological molecules such as proteins, lipids, and ribonucleic acids (RNAs) to their target cells and thus modulate their biological properties and phenotype [13]. In the current review, we will present the existing knowledge about the characteristics and biological properties of EVs in the most frequently encountered hemoglobinopathy worldwide, sickle cell disease.

## 2. Extracellular Vesicles

Thus far, the most well characterized EVs are exosomes, microparticles (also named ectosomes or microvesicles), and apoptotic bodies. Although they differ in their biogenesis pathways, they exhibit some overlap in their physical characteristics, such as their size and density (see bellows). Since numerous studies do not provide any information on the biogenesis of these vesicles and used either their size and/or their density to classify them, the International Society of Extracellular Vesicles has introduced a new classification of these vesicles and classified them as small EVs (sEVs), medium (mEVs), and large EVs (lEVs) [14]. However, in this review, we will use the term of exosomes, microparticles, and apoptotic bodies since they exhibit different biological properties and biogenesis pathways to an extent [15].

### 2.1. Classification and Biogenesis of EVs

#### 2.1.1. Classification of EVs

Exosomes, with a diameter range of 30–150 nm, are secreted from an exocytic chamber called the multivesicular body (MVB). In contrast, microparticles (MPs) exhibit a wider diameter range of 100–1000 nm and are produced from the outward blebbing of the plasma membrane. At last, apoptotic bodies (1.0–5.0 µm) are shed from the cytoplasmic membrane of apoptotic cells. Although EVs are spherical, since they do not have an internal supporting structure similar to cells cytoskeleton, they can appear as cup-shaped structures [16] owing to fixation and dehydration procedures necessary for EVs visualization through transmission electron microscopy. When visualized using atomic force microscopy, they can be deformed during sample preparation and imaging [17]. The latter EVs also exhibit alteration of the phospholipid symmetry with an abnormal externalization of phosphatidylserine, a feature shared with MPs [18]. Table 1 summarizes the main characteristics of exosomes, microparticles, and apoptotic bodies [17,19].

#### 2.1.2. Production of EVs

Exosomes derive from the endosomal system and their biogenesis involves three main steps, namely the intraluminal budding of endosomal compartments, the formation of the intraluminal vesicles (ILVs), and the fusion of MVBs with plasma membrane. It has been shown that ILV formation is under the control of Endosomal Sorting Complexes Required for Transport machinery [20], but the release of exosomes could be regulated by an Endosomal Sorting Complexes Required for Transport-independent manner, which involves tetraspanin microdomains and lipids raft [21,22]. If several mechanisms have been proposed for the release of exosomes [12], the proteins with sorting functions critical for the recruitment of cargo remain largely unknown. Their formation appears to occur both constitutively and in response to various triggers in most cell types, if not all [11,23]. 

In contrast to exosomes and apoptotic bodies, MPs are produced in a few seconds after stimulation [24]. They are formed by regulated release by budding/blebbing of the plasma membrane (Figure 1) and their release is increased in cells submitted to stress conditions, which leads to local cytoskeletal rearrangements and membrane budding [25,26]. Indeed, the increase of intracellular Ca^2+^ concentration induced by these conditions affects the function of three enzymes, namely floppase, scrambase, and flippase, that are involved in the maintenance of the asymmetry of cellular lipid bilayers, and leads to the externalization of phosphatidylserine (PS) [27,28]. PS exposure is believed to be a key event in MP formation. Moreover, the rise of intracellular Ca^2+^ activates proteases that cleave the cytoskeleton, thereby weakening its interaction with plasma membrane and allowing the release of MPs [29]. In addition, several molecules modulating the organization of the cytoskeleton have been shown to either increase or decrease the production of MPs [19]. MPs are usually described as exhibiting PS externalization, although MPs without externalized PS have also been described [30]. Whether this observation results from a lack of sensitivity of the detection method used remains an unanswered question. 

While exosomes and MPs could be secreted during all the cellular life, apoptotic bodies are only produced during programmed cell death. The latest stages of apoptosis are nuclear chromatin condensation followed by membrane blebbing and the destruction of the cellular content into distinct membrane vesicles, the so-called apoptotic bodies [31]. In contrast to the two other types of EV previously described, apoptotic bodies exhibit a permeable membrane [32]. In addition to these large vesicles produced (1000–5000 nm), smaller vesicles are also released [33], but it remains unclear whether the production of these vesicles involves membrane blebbing.

### 2.2. Isolation of EVs

Most technical procedures used to isolate EVs are based either on centrifugation or ultracentrifugation of biological fluids and cell supernatants, for apoptotic bodies or exosomes and MPs, respectively [11]. Ultracentrifugation facilitates later research, but gives a relatively poor sample purity. Another physical method is based on a density gradient. This method provides a high purity but is more complex than ultracentrifugation. Moreover, another method using immunomagnetic beads is less time-consuming than the two previous ones, but requires reagents, which are significantly more expensive [34]. The standardized protocols for exosomes purification may include ultracentrifugation coupled with subsequent sucrose density gradient ultracentrifugation or, alternatively, sucrose cushion centrifugation [16]. In contrast, standard isolation protocols for MPs are still lacking. Since it has been well documented that pre-analytical and analytical conditions significantly impact both quantitative and qualitative MPs analysis [11,35,36,37], specific recommendations and guidelines have been produced [37,38]. However, many groups did not apply these recommendations, leading to confusing and conflicting results. One reason for not following these recommendations is, for example, that a double centrifugation at 2500 g used to discard platelets, also depletes the samples in larger MPs [39]. Indeed, there is an overlap in size between the largest MPs and the smallest platelets [40]. To our knowledge, no standardized protocols have been produced for isolation of apoptotic bodies. It is worth noting that none of the procedures used so far have allowed purification of only one type of EVs. 

Many techniques have been used for quantitative and/or qualitative analysis of EVs, such as Western-blot, flow cytometry, dynamic light scattering, nanoparticles tracking analysis (NTA), scanning and transmission electron microscopy, cryo-electron microscopy, and atomic force microscopy [17,41]. Some of these techniques, NTA for instance, allow researchers to determine the concentration of EVs, but not their cell type-of-origin and their composition. Such techniques can produce results which are challenging to interpret, such as increased EV levels, which can be accounted for by increases in multiple, or sometimes only one EV subtype. Moreover, it seems crucial to determine EVs composition to better interpret increased EV levels, which can result from increased number of cells producing EVs, and/or from increased activation level of some cells. For all these reasons, up to now, flow cytometry is clearly the most used technique for EVs analysis, including for exosomes, using beads conjugated with antibodies targeting specifically proteins overrepresented on their surface. A flow cytometer allows for each event passing through its flow cell, to determine its size, granularity, and fluorescence intensity for several wavelengths. Fluorescence stems from fluorochrome-conjugated antibodies binding specific targets on or inside the EV. Considering a target, when the positive and negative populations are not clearly separated, a Fluorescence Minus One (FMO) control is crucial to set the upper boundary of the background signal. Using fluorescent probes, such as labeled annexin A5, a protein with high affinity for PS, and labeled antibodies binding membrane proteins specific of each blood cell type, plasma concentration and cellular origin of MPs could be theoretically established. However, flow cytometry encounters several shortcomings, including limited sensitivity and resolution, leaving uncharacterized a significant proportion of the smallest MPs, above all with the less sensitive flow cytometers [42]. Clearly, improvements for both isolation and analytic procedures are still needed.

### 2.3. Composition of EVs

EVs are composed of membrane lipids, cytoskeletal, cytosolic, and plasma membrane proteins and may contain several types of RNA including mRNA, miRNA, as well as ncRNA [43]. Genomic DNA has also been detected in apoptotic bodies [44]. Overall, these vesicles contain a large number of molecules with biological activities related to their involvement in cell-to-cell crosstalk. In addition, it has been recently shown that EVs contain biologically active cytokines that could be released from these vesicles to their targeted cells by a yet uncharacterized mechanism [45]. Recent reports also showed that EVs generated *ex vivo* contain functional mitochondria [46,47,48].

Significant attempts have been undertaken to identify the molecular content of the different types of EVs and several public on-line databases have been produced, such as Vesiclepedia (www.microvesicles.org/, accessed on 29 July 2022) [49], Evpedia (www.evpedia.info, accessed on 29 July 2022) [50], and Exocarta (www.exocarta.org, accessed on 29 July 2022) [51]. It is worth noting that proteomic profiles are highly dependent on the procedures used to isolate these EVs and, therefore, leading to uncertainties on their real content [43]. Besides, while protein profiles have been initially thought to allow the identification of EV type, it seems that no single marker can undoubtedly identify EVs. For example, it has been shown that CD63, CD81, and CD9 were not specific markers of exosomes but could be detected in MPs and apoptotic bodies [52,53]. Nevertheless, it has been shown that both cell type origin and triggers leading to the release of EVs significantly impact their content and, therefore, their biological properties [43].

## 3. Pathophysiology of SCD

### 3.1. Physiological Hemostasis and Inflammation

The pathophysiology of SCD relies on the disturbance of several physiological processes, among which coagulation, vasoregulation, and inflammation are crucial. “Hemostasis” comprises all the processes permitting to prevent excessive blood loss following injury, including vasoregulation, which refers to the mechanisms allowing to modulate blood vessels diameter. The term “inflammation” corresponds to the reactions to fight against a pathogen. However, in SCD these reactions can occur in the absence of microorganisms and are associated with hemolysis-mediated release of DAMPs (damage-associated molecular patterns) [54] and MPs, among others. 

#### 3.1.1. Normal Hemostasis

When a blood vessel is severed or punctured, a three-step process occurs to prevent further loss of blood: vascular spasm (a vasoconstriction step to reduce blood losses), platelet plug formation and finally coagulation. As shown in Figure 2A, the coagulation cascade leads to the conversion of prothrombin into its active form called thrombin, and ends to the thrombin-mediated conversion fibrinogen into fibrin, which forms a mesh in which red blood cells and platelets are trapped. The extrinsic pathway, also known as the tissue factor pathway, is initiated due to a trauma undergone by extravascular cells, which provokes the exposure of tissue factor, the coagulation factor III. The intrinsic pathway, also called contact activation pathway, typically begins by the activation of factor XII, when it encounters anionic molecules of the damaged vessel wall. Importantly, both the intrinsic tenase and the prothrombinase complexes assemble on negatively charged phospholipids, it is to say on phosphatidylserine.

Nitric oxide (or NO) is the main vasodilator [55]. It inhibits the action of the most potent vasoconstrictor, endothelin-1 [56], by acting at the transcriptional and translational levels, but also by impeding its release [57]. NO was also shown to inhibit the expression of adhesion molecules by erythrocytes and leukocytes [58], and to prevent platelet aggregation [59]. The production of this vasodilator gas by NO synthase (NOS), from arginase, is stimulated by shear stress, platelet aggregation, and thrombin; whereas hypoxia and some pro-inflammatory cytokines increase endothelin-1 concentration (Figure 2B). However, wall shear stress (WSS), the dragging frictional force generated by blood flow and blood viscosity, is the main physiological NOS stimulus [60]. Moreover, both increase [61] and decrease [62,63] in arterial caliber in response to increases or decreases in WSS, respectively, and have been shown to involve endothelial release of NO. These results highlight the key role of NO, to allow vessels to adapt their diameter variations in WSS.

#### 3.1.2. Normal Inflammation

In case of infection, neutrophils are the first recruited leukocytes. When they encounter a pathogen, they can phagocytose it, release granules containing antibacterial proteins into the extracellular milieu to kill it, or in case of a high activation level, release neutrophil extracellular traps (NETs) to trap the microorganism [64] and facilitate its phagocytosis. The notion that NETs could not only trap, but kill pathogens thanks to their decorating antimicrobial proteins [65], is still a matter of debate. 

Typically, neutrophil recruitment to an infected site requires its tethering, rolling, firm adhesion, crawling, and transmigration to reach the infected site (Figure 2C). Tissue-resident leukocytes release inflammatory mediators to change the endothelium adhesive properties, or endothelial cells can be activated following the detection of pathogens by means of pattern-recognition receptors (PRRs). Therefore, endothelial cells express P-selectin at their membrane, within minutes. P-selectin interaction with neutrophil P-selectin glycoprotein ligand-1 (PSGL-1), allow the tethering (that is, the capture) of the free-circulating leukocyte. Endothelial cell activation also triggers *de novo* synthesis and as such upregulation of E-selectin, within about 90 min [66,67]. E-selectin, which preferentially binds neutrophil L-selectin, projects less further above the endothelial surface of endothelial surface than P-selectin, and have partially overlapping functions with this last protein, allows to slow neutrophil rolling [68]. Additionally, firm adhesion stems from interaction of integrins with intercellular adhesion molecule-1 (ICAM-1) and ICAM-2 endothelial molecules. Lymphocyte function-associated antigen-1 (LFA-1, also called CD11a-CD18) and macrophage-1 antigen (MAC-1, also known as CD11b-CD18) are constitutively expressed by neutrophils but, to allow adhesion they require activation by a combination of mechanisms involving: positively charged chemokines [69], MRP8/14 secretion [70], talin, and kindlin-3 binding to the β chain of LFA-1, which, respectively, also cause conformational changes to further decrease rolling velocity and to allow neutrophil arrest [71]. The leukocyte then begins a MAC-1 dependent crawling step towards the exit site [72]. It is worthwhile noting that neutrophils recruitment in other tissues, where high shear stress is encountered, such as the brain for instance: platelets which can express more P-selectin than endothelial cells, first adhere to the endothelium, and then allow neutrophils recruitment [73]. However, after firm adhesion and crawling, it generally takes 2–5 min for the neutrophils to cross the endothelium, and then 5–15 min to cross the basement membrane [66]. Subsequently, neutrophils are directed by a gradient of chemo-attractants [74]. This review focuses on intravascular adhesive mechanisms, and only briefly deals with extravasation given that this process is less relevant to SCD pathophysiology; but for more specific details readers are encouraged to refer to excellent reviews by Ley et al., 2007, and Vestweber, 2015 [66,70].

### 3.2. Dysregulated Mechanisms in SCD

SCD is a complex, evolutive, and clinically heterogeneous disease. In deoxygenated vascular areas, HbS forms polymers, which makes sickled red blood cells (RBCs) less deformable and more fragile. The resulting hemolysis stimulates the bone marrow and accounts for the elevated count of stress reticulocytes of SCD patients. Whereas the transit time of RBCs in deoxygenated territories should be insufficient to cause their sickling [75]; platelets, neutrophils, and endothelium pro-adhesive phenotypes observed in SCD, may decrease microvascular blood flow, thereby increasing RBC transit time and allowing their sickling before leaving the microcirculation [76]. These sickled RBCs, but also activated neutrophils, platelets, endothelial cells are the main actors in SCD. Therefore, although SCD results from a single point mutation, its pathophysiology relies on the disturbance of several pathways, owing to abnormalities such as elevated hemolysis level and stress undergone by the vascular endothelium.

#### 3.2.1. Pro-Coagulant State

A procoagulant state is one of SCD pathophysiology hallmark [77,78]. SCD hypercoagulable state has been associated with increased risks of pulmonary hypertension [79], in situ thrombosis of small vessels and venous thrombosis [80]. SCD patients exhibit low protein C and S levels, suggesting their chronic consumption due to a constantly activated coagulation cascade [81]. Tissue factor (TF) was reported to be elevated on SCD patients’ monocytes [82], neutrophils [83], and circulating endothelial cells [83]. Heme has been shown to promote TF expression by mononuclear and endothelial cells [84,85]. This is consistent with the constant detection in SCD patients’ plasma, of high levels of coagulation markers such as D-dimers, plasmin-antiplasmin (PAP) complexes, thrombin-antithrombin (TAT) complexes, and prothrombin fragment 1.2 (F1.2) [81]. 

The hyperactivity of the coagulation system in SCD is also caused by reticulocytes or RBCs exhibiting externalized phosphatidylserine (PS). Owing to intravascular and extravascular hemolysis, SCD patients exhibit erythropoiesis expansion and, therefore, increased reticulocytosis. Although PS exposure by immature reticulocytes seems to be normal during hematopoiesis, hyposplenia, or functional asplenia observed in SCD due to abnormal RBCs trapping, increases the count of PS-exposing circulating mature reticulocytes [86]. Moreover, increased intracellular calcium concentration due to sickling and dehydration, but also oxidative stress, account for elevated counts of PS-exposing sickle RBC and reticulocytes [87,88,89]. These cells may promote the activation of the coagulation cascade, since PS is known to provide a docking site for tenase and prothrombinase complexes, which activates the intrinsic pathway (Figure 3A). This correlates with reports of correlations between F 1.2, D-dimers and PAP complexes, and PS-bearing sickle RBCs [90,91]. 

#### 3.2.2. Decreased Nitric Oxide Bioavailability

SCD patients are known for having a 50% increase in cardiac output [92], and a lack of RBC deformability [93]. These two altered parameters contribute to the increased WSS observed by Belhassen et al. in SCD patients [60]. Intriguingly, this augmented level was accompanied by an unchanged vessel diameter, when compared to healthy controls; thereby suggesting an impaired capacity to adjust vessel caliber to WSS in SCD. This failure to adjust arteries diameter could result from defects in the transduction of the shear stress signal, from impairments in the synthesis or the release of NO, or from an accelerated degradation of NO. The results of this group excluded the two first hypothesis, in the favor of the last one. Consistently, NO bioavailability is known to be drastically reduced in SCD owing to the elevated hemolytic rate [94]. Indeed, hemolysis allows the release of arginase, which impedes NO production by using L-arginine to produce ornithine (Figure 3B). Hemolysis also induces the release of hemoglobin in the plasma, which reacts with NO to form methemoglobin and nitrate. Consistently, decreased L-arginine concentration, coinciding with high arginase plasma levels, were reported in SCD patients [95], but also elevated concentrations of free hemoglobin and methemoglobin [95,96], which all account for a decreased NO bioavailability [97] and, therefore, a reduced WSS-mediated vasodilation. The decreased bioavailability of NO and the resulting endothelial dysfunction have been associated with an increased risk of pulmonary hypertension [98], legs ulcers [99], stroke [100] and priapism [101]. Besides, NO being an inhibitor of endothelin-1 (ET-1), the decrease of its bioavailability accounts for the high level of ET-1 observed in SCD. ET-1 binding to its receptor, has been shown to increase calcium concentration inside erythrocytes and to facilitate Gardos channel opening, thereby potentiating erythrocyte dehydration and so increasing HbS concentration and its propensity to polymerize [102]. Moreover, since NO is an inhibitor of the expression of adhesive proteins by the vascular endothelium [58], NO-scavenging by free heme and arginase-mediated decrease in NO production both have a role in the pro-adhesive phenotype of endothelial cells in SCD.

#### 3.2.3. Pro-Inflammatory State

Blood flow obstruction during VOCs, causes severe pain, and repeated VOCs can lead to organ failure. A substantial proportion of the knowledge regarding the adhesive processes leading to VOCs was acquired thanks to murine models of the disease [103,104] or to microfluidics [105]. The etiology of the main SCD-associated complication, VOC, involves the capture of neutrophils by the activated endothelium. Upon rolling, these neutrophils exhibit active integrin that allow their full arrest. This adhesion in post-capillary venules causes trapping of platelets, other neutrophils, and above all RBCs, which lead to the occurrence of a vascular occlusion [106]. The key role of neutrophils is highlighted by the absolute contraindication of myeloid growth factors such as granulocyte macrophage colony-stimulating factor (GM-CSF) or granulocyte colony-stimulating factor (G-CSF) in SCD patients [107,108,109,110]. Aged neutrophils [111], which are overrepresented in SCA patients [112], present a 70% increase in the adhesive molecule MAC-1 [113]. This last integrin appears to have a crucial role in SCD since it allows neutrophils to adhere to endothelial ICAM-1, but also to capture circulating RBCs [76] (Figure 3C).

The RBC lifespan, which is normally of 120 days, is reduced in SCD to about 12 days. Hemolysis favors the release from the bone marrow, of reticulocytes; reaching a tenfold increase in their count, compared to normal conditions. These reticulocytes express high levels of adhesion molecules, among which intercellular molecule-4 (ICAM-4), a molecule shown to bind to MAC-1 and so allow RBC-neutrophil interactions [114]. Sickle patient RBCs commonly exhibit externalized PS, what promotes their adhesion [87]. Sickle red cells also exhibit alterations leading to the abnormal activation of erythroid adhesion molecules such as Lu/BCAM, ICAM-4 and CD44 [115,116]. As a consequence, erythrocytes interactions with the endothelium or with circulating of adherent platelets and neutrophils are facilitated [104,117,118] (Figure 3C). In addition, the pro-adhesive phenotype of RBCs of SCD patients and the reduced vasodilatory capacity account for an increased number of interactions between RBCs and endothelial cells, which was shown to upregulate the expression of VCAM-1 and ICAM-1 genes [119].

Besides, enhanced oxidative stress, partly due to HbS auto-oxidation, which induces superoxide anion, hydrogen peroxide, and hydroxyl radical production [120,121], is associated with vascular alterations in SCD patients [122]. Free heme promotes the secretion of pro-inflammatory cytokines by activating monocytes/macrophages [123], platelets [124], endothelial cells [84], and neutrophils [125]. Moreover, NETs released by the latter cell type, were detected at high concentration in the plasma of SCD patients at steady state, with a further increase during crisis [125,126]. 

### 3.3. EVs as Novel Biomarkers in SCD

Circulating EV concentration has been shown to be increased in several cardiovascular diseases [5,127]. Since plasma EVs concentration and composition reflects specific signatures of cellular activation and injury, EVs characteristics may represent in the future, a useful diagnostic and prognostic tool in several diseases.

In SCD, the concentration of the two most commonly identified MPs subtypes, RBC-, and platelet-MPs is increased, compared to healthy controls [128,129] (Table 2). HU treatment impact on MPs concentration is controversial, since several reports showed decreases [128,129,130], unchanged [14,131], or increased [132,133] levels. These conflicting results could be accounted for by the large interindividual variation in MPs concentration in SCD. However, a longitudinal study reported no change in MPs concentration in patients receiving HU for 24 months [134]. To further characterize the biomarker status of MPs, an observational study with an estimated enrollment of 360 participants has also been initiated (NCT012422878). A positive history of osteonecrosis of the femoral head [135], leg ulcers [136], acute chest syndrome, and pulmonary hypertension [129] has been associated with elevated concentration of MPs from various cell types. During VOC, the concentration of PLT- and RBC-MPs was also reported to be increased in cross-sectional settings [132,137,138], and in longitudinal studies including 17 SCD patients [139] or 32 SCA patients [140]. Our group reported that SCA patients with frequent VOCs had increased levels of PLT-MPs, compared to SCA patients with rare crises [141]. In SCA patients, we showed using two longitudinal cohorts, that circulating PLT- and RBC-MPs PS exposure level was increased during VOC, but decreased after two years under HU treatment, when compared to steady-state conditions [39,134,140]. Therefore, PS exposure by these MPs subtypes seems to be a promising marker of clinical severity and of HU treatment efficacy. Further studies based on large cohorts are warranted to determine if MPs PS exposure is a prognosis marker. Moreover, it seems important to determine if the cost effectiveness of such a test is favorable, above all for the medical monitoring of patients having frequent VOCs or patients who cannot be treated with HU. Moreover, the 2-year long HU treatment provoked an increase in the size of RBC-MPs, probably resulting from the improved RBC hydration provided by this drug [134].

**Table 2 bioengineering-09-00439-t002:** Studies reporting increased EV levels in SCD. Markers proving an origin from RBCs, reticulocytes, platelets, monocytes, endothelial cells, leukocytes, or progenitor cells are mainly CD235a, CD71, CD41a, CD14, CD106, CD45, or CD309/CD34 respectively.

Reference (Number of Included Patients)	Method	EV Type	EV Cell Type-of-Origin	EVs Concentration
Dembélé et al. [136] (232 SCA patients)	Flow cytometry	MPs	RBCs, platelets, monocytes, endothelial cells, progenitor cells	RBC-MPs/mL: 6678 (SCA), 1533 (Controls); PLT-MPs/µL: 3320 (SCA), 2627 (Controls)
Kasar et al. [138] (45 SCD patients)	Flow cytometry	MPs	RBCs, platelets, endothelial cells, monocytes	RBC-MPs (events/µL): 7.59 (SCD), 0.10 (Controls); PLT-MPs (events/µL): 12.58 (SCD), 1.59 (Controls)
Shet et al. [137] (16 SCD patients)	Flow cytometry	MPs	RBCs, platelets, monocytes	RBC-MPs/µL: ~650 (SCD), ~30 (Controls); PLT-MPs/µL: ~50 (SCD), ~50 (Controls)
Gerotziafas et al. [142] (92 SCA patients)	Flow cytometry	MPs	RBCs, platelets	RBC-MPs/µL: 1370 (SCA), 69 (Controls); PLT-MPs/µL: 1897 (SCA), 752 (Controls)
Garnier et al. [143] (33 SCD patients)	Flow cytometry	MPs	RBCs, platelets, monocytes, endothelial cells, leukocytes	RBC-MPs/µL: 631 (SCA), 260 (HbSC); PLT-MPs/µL: 6485 (SCA), 4014 (HbSC)
Lappin-carr et al. [144] (33 SCD patients)	Imaging flow cytometry	Exosomes	RBCs, endothelial cells, hematopoietic progenitors, lymphocytes, monocytes, platelets	RBC-Exo/µL: 31,338 (SCD), 9661 (Controls); PLT-Exo/µL: 2702 (SCD), 1116 (Controls)
Khalyfa et al. [145] (32 SCA patients)	Imaging flow cytometry, electron microscopy	Exosomes	Endothelial cells, endothelial progenitor cells, monocytes, platelets, RBCs	RBC-Exo/µL: 2,760,753 (SCA), 1,768,125 (Controls); PLT-Exo/µL: 5653 (SCA), 5435 (Controls)

Unlike for MPs, only a few reports regarding exosomes in SCD have been published, and none dealing with apoptotic bodies. We collaborated with a group who showed that circulating exosome concentration was increased in SCD patients, compared to the controls [144,145]. They also reported associations between the severity of the disease and the counts of exosomes produced by monocytes, lymphocytes, and endothelial cells [145]. Of note, the same group reported a signature of microRNAs contained into circulating exosomes, which distinguished severe from mild clinical profile between SCA patients [145]. These results suggest that circulating exosomes could become useful diagnostic and prognostic tools used in clinical settings.

### 3.4. Effects of EVs in SCD

The externalization of PS is a key feature of MPs. Indeed, as stated before, MPs PS provides a docking site for the intrinsic tenase and the prothrombinase complex [146,147,148]. In line with the pro-coagulant role of MPs due to their PS and TF [137], Scott syndrome, characterized by a defect in platelet-derived MPs production, has been associated with increased bleeding [149]. MPs generated *ex vivo* by platelets or erythrocytes, have also been shown to trigger thrombin generation via factor XIIa [150]. However, whether circulating MPs have similar biological properties remains unknown. Moreover, although PLT-MPs are known to represent the commonest MPs subtypes in the circulation, most studies in SCD, report associations between the concentration of RBC-MPs and coagulation cascade activation. This paradox may be explained by the higher exposure of PS by RBC-MPs, compared to PLT-MPs observed at steady state, in HU-treated or untreated patients, and even during crisis [39,143].

PS was shown to allow MPs binding to endothelial cells [151,152]. Consistently, increased PS exposure was associated to increased fusion with these cells [39,151,153]. These results are supported by the report of the expression of a PS receptor (PSR) by endothelial cells [85], thereby allowing MPs to bind to these cells, and to modify their phenotype. Moreover, PS was shown to allow MPs to retain heme, which was hereafter transferred to endothelial cells [154]. RBC-MPs were shown to be internalized by myeloid cells and to promote inflammatory cytokine secretion along with adhesion to endothelial cells [155]. Barry and colleagues reported that PLT-MPs induced endothelial ICAM-1 expression [156], and Wang et al. showed that monocyte-derived MPs increased ICAM-1, VCAM-1, and E-selectin expression, also in endothelial cells [157]. These results may partly explain why the infusion of MPs was shown to trigger vaso-occlusion mice kidney [154]. Since the content of MPs is known to be influenced by the stimuli triggering their production and by their cell type of origin, our group used circulating MPs, instead of MPs generated *ex vivo*. Our results showed that MPs circulating during VOC, triggered a PS-dependent ICAM-1 overexpression, compared to MPs from the same patients but at steady state [39]. On the contrary, ICAM-1 expression was reduced when MPs were isolated from the plasma of SCA patients under HU treatment (Figure 4). Moreover, the adhesion of SCD neutrophils to MPs-stimulated endothelial cells was decreased when using MPs from HU-treated patients, and increased in an ICAM-1 dependent manner using MPs from patients in VOC. We also showed that RBC-MPs from SCA patients at steady state, increased ICAM-1 expression and cytokines production in a TLR-4-dependent manner, compared to MP from healthy controls [158].

Exosomes generated *ex vivo* by mesenchymal stem cells, multipotent progenitors found in various tissues and having tissue-repair functions, were recently shown to have procoagulant activities thanks to their PS and TF [159]. If such exosomes are found in sufficient levels in the blood of SCD patients, they could play a crucial procoagulant role in this disease. Vats et al. showed that pretreatment of platelets with LPS induced inflammasome activation and the production of EVs richly packaged with IL-1β [160]. These EVs had a size corresponding to the one of exosomes, between 50 and 100µm for most of them. Injection of such EVs from SCD platelets, was sufficient to induce lung vaso-occlusion in SCD mice. Moreover, this deleterious effect of platelet-derived exosomes was reduced using an IL-1 receptor antagonist. Their results suggest that drugs preventing platelet exosomes production may be of benefit in SCD. Contrary to the previous group, which used exosomes generated *ex vivo*, another one used circulating exosomes. The mode of the size distribution curve for their EVs was 95nm, and these EVs were rich in exosomal proteins (CD63 and flottilin-1). This group showed that SCD exosomes cause endothelial monolayer disruption [144,145,161]. Importantly, the extent of the endothelial disruption was even greater using exosomes circulating during acute chest syndrome [161] or during VOC [162], compared to exosomes purified from the plasma of patients at steady state.

## 4. Conclusions

MPs concentration and PS exposure have been repeatedly shown to be modulated according to clinical conditions such as HU treatment or crisis. However, the content of circulating MPs has been poorly studied and should be further addressed to better understand MPs biological properties. Although they are less studied, exosomes of SCD patients may also become useful biomarkers given that their count and the microRNAs they contain are associated to the severity of the disease. EVs were shown to carry cytokines in several clinical conditions [45]. This pool of encapsulated cytokines should also be studied as it modulates cell phenotype, such as plasma cytokines do. Furthermore, given that the size, the content and the properties of EVs are known to fluctuate according to the factors provoking their release [157,163,164], more effort should be made to replicate mechanistic discoveries obtained with EVs generated *ex vivo*, using circulating EVs.

Circulating SCA MPs have been shown to trigger a PS- and ICAM-1-dependent increase in neutrophil adhesion [39], the first step in the mechanisms leading to vaso-occlusion. Since MAC-1 and LFA-1 complete blockade can be dangerous [165], given that these integrins are crucial in diapedesis; other strategies based on a MPs PS-blocker such as annexin A5, or targeting selectins [106], could be helpful by reducing neutrophil stasis in post-capillary venules.

Altogether, the studies cited in this manuscript show that EVs are biomarkers and actors in SCD, they illustrate the dramatic increase in the knowledge acquired in this field since EVs discovery in 1967 [10], thereby raising even more interest for future advancements needed to better fight against SCD.

## Figures and Tables

**Figure 1 bioengineering-09-00439-f001:**
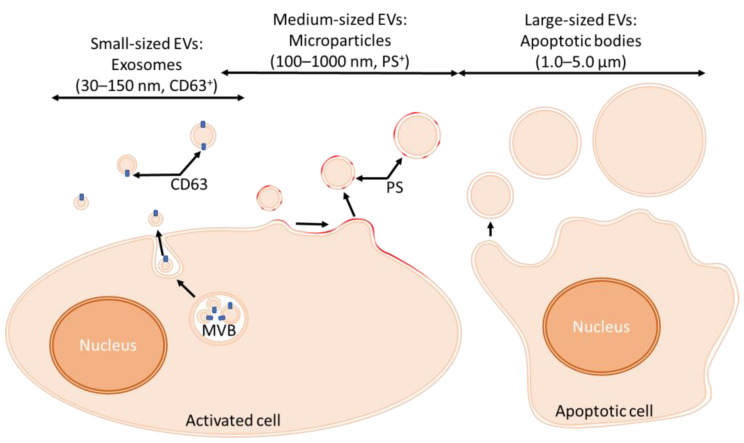
Biogenesis of EVs. Exosomes are produced following the fusion of multivesicular bodies (MVB) with plasma membrane. Common exosomal markers are CD63, CD9, CD81, and flottilin. Microparticle production results from intracellular Ca^2+^ concentration increase. These medium-sized EVs expose phosphatidylserine (PS). Apoptotic bodies are produced during apoptosis and are the larger type of EVs.

**Figure 2 bioengineering-09-00439-f002:**
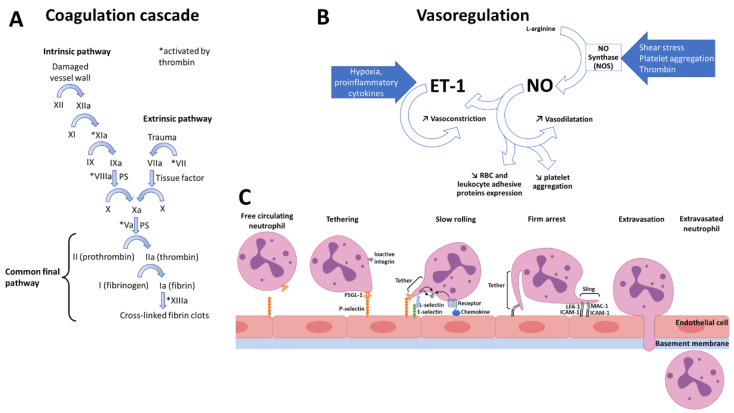
Physiological coagulation cascade activation, vasoregulation, and diapedesis. (**A**) Coagulation cascade relies on the activation several factors and is composed of two pathways, the intrinsic, and the extrinsic one, which both lead to the common final pathway; (**B**) Vasoregulation is mainly regulated by the balance between the vasoconstrictor endothelin-1 (ET-1), and the vasodilator nitric oxide (NO), which inhibits ET-1; (**C**) During diapedesis, free circulating neutrophil are tethered, then they start rolling slower and slower due to their tethers and slings, until their firm arrest. Hereafter they crawl to find the exit location where they are extravasated.

**Figure 3 bioengineering-09-00439-f003:**
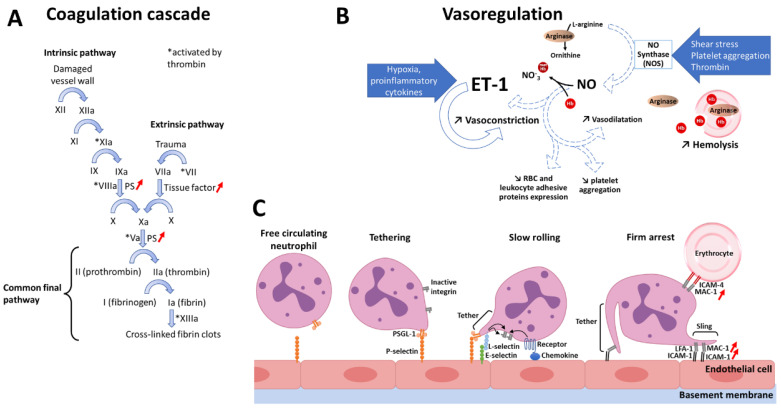
Pro-coagulant and pro-inflammatory context in SCD. (**A**) In SCD, phosphatidylserine (PS) and tissue factor (TF) are exposed at elevated levels by several cell types and microparticles, which contributes to the increased activation state of this coagulation cascade; (**B**) The high hemolysis rate observed in SCD, contributes to the loss of vasodilatory reserve reported in this disease. Indeed, following hemolysis, free arginase will deplete the substrate allowing to form nitric oxide (NO), and free hemoglobin will carry out NO-scavenging. Therefore, the level of the vasoconstrictor endothelin-1 (ET-1) is increased; (**C**) Several mechanisms including increased interactions of erythrocytes with endothelial cells, allow the overexpression of intercellular adhesion molecule-1 (ICAM-1). Moreover, SCD neutrophils have been shown to overexpress the integrin macrophage MAC-1, thereby allowing increased interaction with the vascular endothelium and with erythrocytes.

**Figure 4 bioengineering-09-00439-f004:**
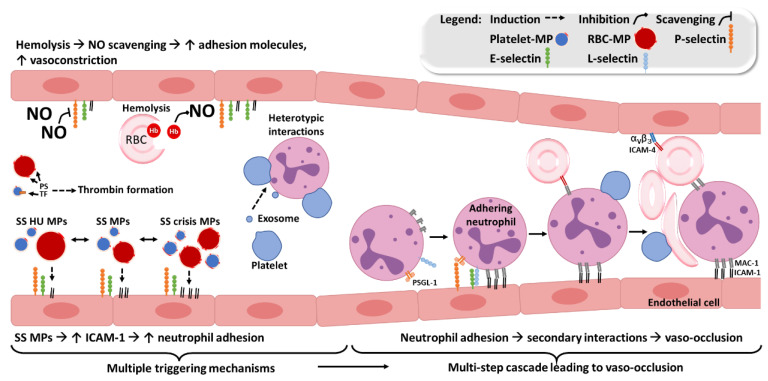
Actors contributing to the occurrence of vaso-occlusions in SCA. Multiple triggering mechanisms lead to a pro-coagulant, a pro-adhesive, and a vasoconstrictive state in SCD. They are partly due to phosphatidylserine (PS) and tissue factor (TF) increased expression by MP, to MPs-mediated increase of endothelial intercellular molecule-1 (ICAM-1) expression and to nitric oxide (NO) decreased bioavailability, respectively. In this context, neutrophils can abnormally adhere to the activated vascular endothelium. Next, secondary interactions with erythrocytes and platelets can lead to a vaso-occlusion, a key event leading to vaso-occlusive crises (VOCs).

**Table 1 bioengineering-09-00439-t001:** Main characteristics of exosomes, microparticles, and apoptotic bodies.

	Exosome	Microparticle	Apoptotic Bodies
Size (nm)	30–150	100–1000	1000–5000
Density (g/cm^3^)	1.13–1.19	1.04–1.07	1.16–1.28
Origin	Multivesicular body	Plasma membrane	Plasma membrane
Formation mechanism	Exocytosis of MVB	Budding from PM	Budding from PM
Production pathway	ESCRT-dependent *	Ca^2+^-dependent	Apoptosis-related pathways

MBV: multivesicular body; ESCRT: endosomal sorting complexes required for transport; PM: plasma membrane; *: ESCRT-independent pathways have also been described.

## Data Availability

Not applicable.

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
