# Peer review of "Extracellular Vesicles in Sickle Cell Disease: A Promising Tool"

_bioengineering, 2022, doi:10.3390/bioengineering9090439_

Round 1

Reviewer 1 Report

The review is devoted to new aspects of the pathogenesis of sickle cell disease with an emphasis on the role of extracellular vesicles. The review systematizes new knowledge in this area and is of interest to a wide audience dealing with sickle cell disease.

According to the reviewer, the manuscript needs to be improved, as well as reduced in terms of well-known facts.

It should be clarified that exosomes display a cup-like shape with certain fixation methods for transmission electron microscopy, while they themselves are spherical in shape.

Please provide a references for the data presented in Table 1.

When describing the composition of extracellular vesicles, it is important to note the potential possibility of containing organelles, for example, mitochondria, as shown by recent studies (DOI: 10.3390/ijms23137408, DOI: 10.1371/journal.pbio.3001166).

In the sections “3.1.1 Normal hemostasis” and “3.1.2 Normal inflammation” the well-known information from the “textbooks” is presented and it overloads the review. These sections are recommended to be excluded from the manuscript.

Table 2 would be more informative if it indicates the specific concentrations of extracellular vesicles, as well as the method of assessment. It is necessary to indicate whether changes in the linear dimensions of extracellular vesicles were studied in these works, which may have additional diagnostic significance.

It is known that extracellular vesicles originated from various cell types, including multipotent mesenchymal stromal cells, have a procoagulant effect due to the expression of tissue factor and the exposure of phosphatidyl serine  on their surface (DOI: 10.3390/cells8030258). Since these mechanisms may be part of the pathogenetic mechanism of SCD, the authors need to consider these mechanisms in more detail.

According to the authors, such an marker as an increase in the number of extracellular vesicles can be a useful for the diagnosis of SCD? An increase in the number of extracellular vesicles is a non-specific marker, and this phenom is founded for various pathologies. Perhaps more correctly, as knowledge accumulates, it would be to determine the pattern of certain molecules contained in the extracellular vesicles of SCD patients. The issue of the value of determining only the quantitative content of extracellular vesicles in the blood requires an extended discussion.

Reviewer 2 Report

The authors subitted a review article in which they evaluated a role of EVs in sicke cell disease. The aim of the study is clear and concise. The article is well organised and has a logical structure. The subsections cover all aspects of the initial hypothesis and ends an informative conclusion. The tables and figures are clear and legible. Overall, the article is impressive and I congratulate the authors on it. However, I would like to put forward several comments to discuss.

1. Practica aspect of measuring EV concentration with flow cytometry requires FMO control. The authors should add a brief description of it.

2. Economic aspect of the biomarke. The author should report their opinion whether the economic component of the biomarker evaluation is lesser that potential hafm from underscoring risk of the patients.

Round 2

Reviewer 1 Report

The correction and the new comments added to the work have substantially improved its quality. I have no further questions.

Author Response

We thank the reveiewer for his/her answer.